# Meeting Cyber Age Needs for Governance in a Changing Global Order

**Oran R. Young [1,]\*** 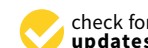**, Jian Yang [2] and Dan Guttman [3,4,5]**

1   Bren School of Environmental Science and Management, University of California, Santa Barbara, CA 93106, USA
2   Shanghai Institutes for International Studies, Shanghai 200233, China; yangjian@siis.org.cn
3   Law School, Tianjin University & Institute for Global Public Policy, Tianjin 300372, China; djguttman@aol.com
4   Institute for Global Public Policy, Fudan University, Shanghai 200433, China
5   US Asia Law Institute, New York University Law School, New York, NY 10012, USA
\*   Correspondence: oran.young@gmail.com

**Abstract:** The advent of the cyber age has created a world in which digital systems, operating on their own and interacting with more conventional material or physical systems, have become an increasingly prominent feature of the landscape of human affairs. This development, affecting every aspect of human life, has generated a class of increasingly critical needs for governance that are difficult to address effectively within the confines of the current global order in which sovereign states compete to maximize their influence in the absence of any overarching public authority. These needs include concerns associated with the management of powerful digital technologies (e.g., artificial intelligence, robotics, machine learning, blockchain technology, the internet of things, and big data) as well as problems relating to the use of these technologies by many actors to exercise influence from the level of the individual (e.g., identity theft) to the level of international society (e.g., foreign interventions in national electoral systems). The challenge of meeting these needs prompts an analysis of processes leading to change in the prevailing global order, energized at least in part by the growing role of the digital systems of the cyber age. Our analysis includes both Western perspectives highlighting changes in the identity and behavior of key actors and Chinese perspectives emphasizing the spread of social narratives embedded in the concepts of *tianxia* and *gongsheng*. While it is premature to make explicit predictions, we conclude with some observations about the most important trends to watch regarding efforts to meet cyber age needs for governance, and we note the connections between these developments and the overarching challenge of fulfilling the suite of goals commonly associated with the idea of sustainable development.

**Keywords:** cyber age; global order; facilitated reform; managed transformation; *tianxia*; *gongsheng*; sustainable development

## 1. Global Order Revisited

An intense debate has arisen regarding the future of the postwar global order, characterized variously as the liberal international order, the rules-based international order, or simply the American world order. For the most part, this debate treats territorially delimited states as the principal actors in international society and directs attention to geopolitical interactions among the leading states. Is the influence of the United States, widely regarded as the principal architect of the postwar order, now receding [1]? Are we witnessing the emergence of a new bipolar or multipolar system that will be more or less stable and sustainable than its predecessor [2]? Will the rise of China trigger the

Thucydides trap, making violent conflict between the United States as an aging hegemon and China as a rising challenger inevitable [3]? How should we assimilate new sources of power or influence (e.g., those associated with rapid developments in information technology) into our efforts to assess the capacity of states to exercise power [4]?

Whatever the merits of individual contributions to this debate, we argue in this article that the debate itself deflects attention from more fundamental developments that are not only generating unprecedented needs for governance on a global scale but also raising fundamental questions about the usefulness of analyses based on the assumption that the world will continue during the foreseeable future to be organized as a society of place-based sovereign states competing to maximize relative gains in the absence of any overarching public authority.

Rising human populations, increased affluence, and acceleration in the pace of technological innovation have combined to turn the Earth into a human-dominated system [5,6]. The result is a hyper-connected earth system characterized by powerful telecoupling among developments occurring in widely separated locations, rapid and often non-linear processes of change, and recurrent surprises giving rise to new and unforeseen needs for governance [7]. This combination of conditions has led thoughtful observers to conclude that the Earth has moved into a new era featuring novel challenges regarding the pursuit of sustainability.

We characterize this new era as the *cyber age* to draw attention to the expanding role of cyberspace and the fact that virtual reality now rivals physical or material reality in many domains. We develop the thesis that the onset of the cyber age has compromised the capacity of states to respond to pressing needs for governance and brought with it or intensified a set of prominent issues that differ from the traditional concerns of world politics in ways that make them difficult—often impossible—to address effectively using familiar tools of public policy.

Several issues of this sort have been on the horizon for some time. A prominent example involves efforts on the part of states to regulate information technologies allowing citizens to become aware of developments occurring elsewhere on the planet in real time and in great detail. The government of China, for example, has invested heavily, with mixed results, in efforts to build a firewall to block or suppress the spread of information about major issues among its citizens. Other issues of a similar nature have begun to surface more recently. A prominent case centers on the deployment of 5G technology. The United States, for example, has sought, with limited success, to pressure allies to avoid using the products of the Chinese multinational Huawei on the grounds that such actions could threaten the security of the United States and its allies.

What issues of the this sort have in common is that they are difficult to address in terms of mainstream approaches to governance in the prevailing global order. Our goal is to initiate a dialogue about innovative ways to address this growing collection of issues and, in the process, to stimulate new thinking about the future of the global order. We do not adopt the naive view that the state-centric order will simply fade away in the coming years to be replaced in a gradual and peaceful manner by some alternative form of order better equipped to tackle the issues of the cyber age. But neither do we accept the view that the state-centric global order is an immutable fact of life.

Social institutions, including macro-political institutions like those that make up the state-centric global order, do change over time. The fundamental narratives that underpin human affairs are social constructs subject to recasting, especially in a setting in which cyber age technologies allow new ways of thinking to spread globally with remarkable speed. We seek to stimulate innovative thinking about such processes rather than to push for premature closure regarding the character of the institutions needed to address new needs for governance. What is called for, at this stage, is a concerted effort to redirect attention from the ins and outs of neo-realist debates about geopolitical interactions to the prospects for operating successfully and sustainably in a substantially altered global order.

## 2. The Cyber Age: A World Transformed

Most of us are comfortable thinking about a world of physical or material spaces, including biophysical systems and social systems that interact with one another in familiar ways. Biophysical systems provide the context within which social systems have evolved. Human communities adapt to the meteorological conditions and natural endowments of their surroundings. Today, we are witnessing the growth of human domination of major biophysical systems. In the case of climate change, to take a prominent example, human activities are increasing concentrations of greenhouse gases in the Earth's atmosphere, triggering far-reaching changes in the climate system. Developments of this sort have given rise to what many now characterize as the Anthropocene, a new era marked by the breadth and depth of the impacts of human activities on biophysical as well as social systems on a global scale [8].

Alongside these developments, we have experienced the rise of digital systems making use of electronic technologies to generate, store, and process data. The result is the creation of cyberspace in which participants interact with one another in a manner that gives rise to virtual reality in contrast to physical or material reality. A prominent case involves the development of computers and the use of the internet to transmit data from one computer to another in electronic forms. Increasingly, digital systems interact with material systems in transformative ways. Figure 1 provides a visual representation of this new world featuring both the growth of cyberspace and interactions among material (biophysical and social) systems and digital systems.

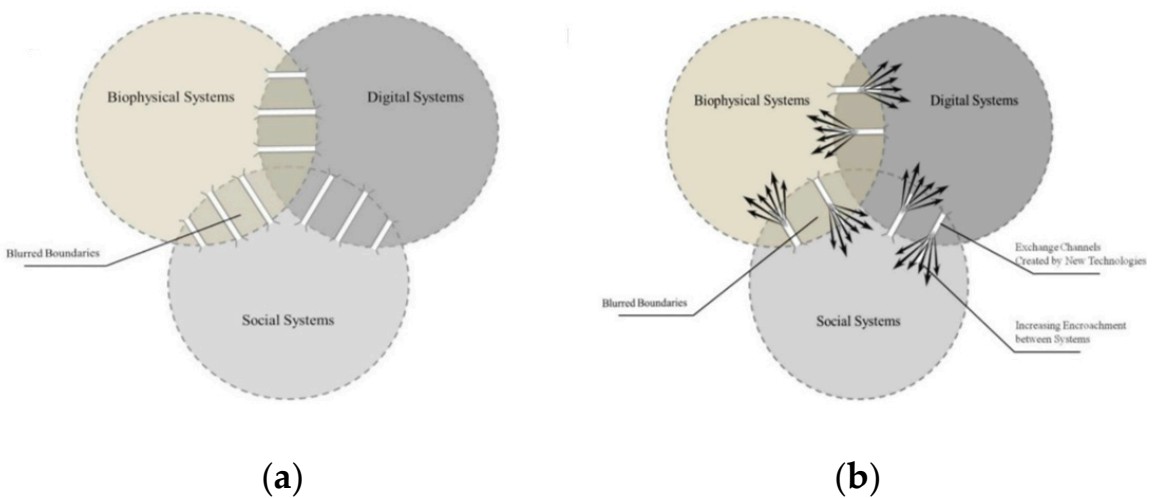

**(a)**　　　　　　　　　　　　　　　　　　**(b)**

**Figure 1.** (**a**) Interacting (biophysical and social) material systems and digital systems. (**b**) Intensification of interactions over time.

In the cyber age, humans have devised technical systems with special functions (such as GPS technology, digital radar technology, airspace control technology, climate monitoring technology) that allow for the development of complex networks in cyberspace in contrast to material space. The degree of openness among the resultant systems has increased; flows of information, materials, and energy among these systems have expanded. The boundaries separating individual systems have blurred resulting in increased uncertainty. Surprises, arising from non-linear and often sudden changes, have become more common. Increased openness for exchange among the collection of systems has led to the growth of complexity, adding new dimensions to the pursuit of sustainability. Governance in the cyber age poses fundamental questions about the locus of authority to deal with interactions among systems, the timing of interventions, and the nature of the tools needed to steer these interactions effectively. Conventional approaches to governance, which assume clear-cut boundaries among systems, are becoming part of the problem rather than sources of solutions. New approaches, designed with a clear understanding of the key features of the cyber age, are required [9,10].

The emergence of cyberspace and the rise of virtual reality began in the decades following World War II with the introduction of semiconductors and transistors making possible rapid electronic transmission of large quantities of data [11]. In the ensuing years, the role of digital systems has grown at an accelerating pace (see Figure 2). Starting with the development of computers and the creation of the internet and the World Wide Web, this development has expanded to encompass various forms of social media, the production of big data, the invention of the Internet of Things, and the introduction of increasingly powerful forms of artificial intelligence. What began with the invention of seemingly esoteric devices by engineers specializing in electronics has produced an array of technologies affecting every aspect of our lives. The end of this set of developments is not in sight. The result has transformed our world not only by revolutionizing interactions among human beings but also by drastically altering human interactions with biophysical systems.

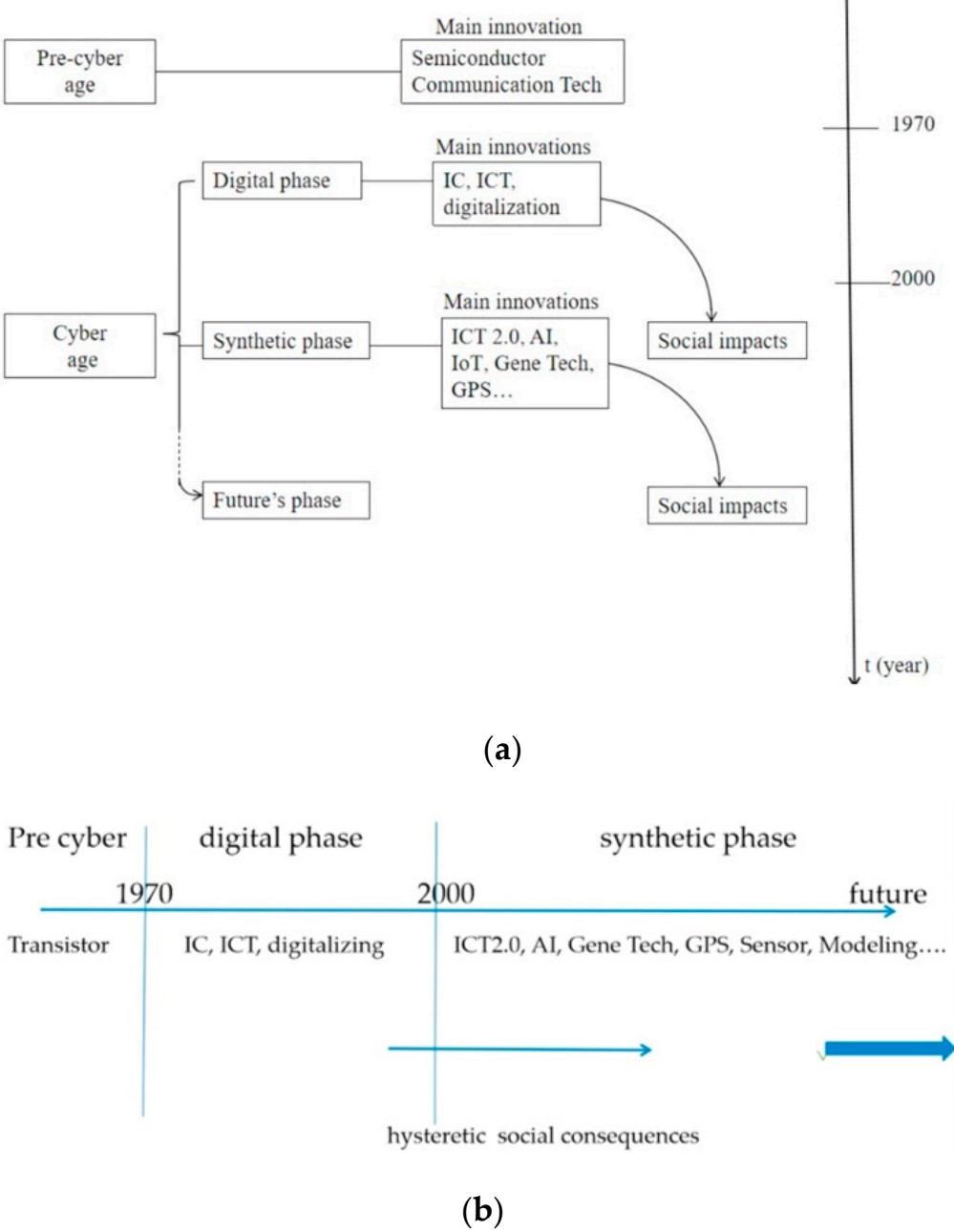

**Figure 2.** (**a**) Main innovations of the cyber age. (**b**) Phases of the cyber age.

The onset of the cyber age is about creating new spaces rather than about discovering existing but previously unknown spaces (e.g., new lands). It centers on uses of cyberspace to distribute all kinds of information content. The result is the creation of virtual systems that interweave with material (biophysical and social) systems in a complex manner. The introduction of digital technology launched the first phase of this new age. During this phase, all contents (text, audio, video) were digitized using mutually translatable coding systems and technical standards. In addition, the continuous development of new digital technologies, dealing with computing, storage, coding, program control and frequency division, has made it possible to achieve global information connectivity through the development of worldwide digital technology infrastructures.

Today, we are experiencing the social impacts associated with the invention of digital technologies. Prominent examples include the widespread accessibility of knowledge regarding sophisticated technologies (e.g., nuclear technology, biotechnology), the growing capacity of an array of non-state actors and even individuals to exercise influence on a global scale, the associated flattening of social structures, and the consequent reduction in the ability of traditional public authorities to control social interactions. The globalization of information and the resultant shifts in values are producing telecouplings among trade and financial systems, subnational levels of government, and social movements that states are unable to manage effectively [12,13].

The introduction of new technologies is blurring boundaries between material (biophysical and social) systems and digital systems and leading to increased interactions among them [14]. These developments expand the scope of cyberspace; they also raise the prospect of passing thresholds or tipping points unintentionally, triggering profound changes in linked systems taking the forms of explosions, cascades, and inflections [7]. This increases the importance among those concerned with resilience or sustainability of thinking about maintaining a safe operating space for humanity not only with respect to biophysical systems (e.g., managing climate change, controlling pandemics) but also with respect to social systems (e.g., creating resilient economies, securing human well-being) [15,16].

Digital technologies enhance human learning ability in some ways. However, our ability to comprehend the dynamics of complex systems remains limited. One response is to improve and rely increasingly on artificial intelligence in an effort to respond to needs for governance arising in the cyber age. However, the resultant improvements in technological capacity give rise to additional and more complex needs for governance. This sets off feedback processes between the creation of new technologies (e.g., 5G, the Internet of Things, machine learning) and the emergence of new needs for governance.

A result of these technological advances is the development of increased autonomy on the part of machines, allowing machines to exchange information with one another and to learn in the absence of human direction. Allowing artificial intelligence to manage systems that are too complex for humans to understand, let alone to manage effectively, has obvious attractions in a world featuring the rapid growth of complexity. However, this response brings with it new dangers if the programs designed initially by human engineers are defective or if the machines are vulnerable to various forms of cyberterrorism, for example, critical systems may crash, undermining resilience and causing irreparable harm to humankind [17].

As might be expected, some human actors approach the onset of the cyber age as a matter of identifying ways to use the resources of cyberspace to generate wealth or power. Among these resources are those involving time, data, knowledge, and ICT infrastructure [11]. It is helpful to divide these resources into two categories: soft space resources and hard space resources. From a soft space perspective, cyberspace provides a place for preaching as with a church, and for trading as with a market. In terms of hard space, cyberspace features bandwidth resources, spectrum resources, and cyber-network address resources. These are all scarce resources under current technological conditions, a fact that provides owners or users with an incentive to exercise control over them to augment their power and wealth.

At the same time, the emergence of cyber-material systems resulting from the overlapping and interpenetration of biophysical, social, and digital systems presents problems for a political order based on the idea of the sovereignty of place-based states. There are no clear answers to questions about both the right and the capacity to exercise sovereignty in cyberspace. States endeavor to assert ownership and the right to control the content and disposition of information in cyberspace. Some non-state actors contest efforts to expand the jurisdiction of states in this realm [18]. However, more fundamentally, the flow of content (text, audio, video) can be separated from the relevant hardware. The ownership of infrastructure provides no guarantee of the ability to control the production and dissemination of content.

In the past, struggles over the control of resources have focused on the establishment and defense of borders. Some actors try to break existing boundaries to obtain new resources; others try to strengthen boundaries in order to control these resources. In cyberspace, the important resources are highly mobile. Multinational corporations now seek to develop and utilize virtual systems to profit from the content of data resources around the world. Whereas material spaces are based on geographical relationships, virtual spaces are based on coding systems and logical circuits. Coding systems regulate human behavior in cyberspace, affecting the social order of cyberspace. Coding systems also act as a resource boundary in cyberspace. The higher the level of coding, the more difficult the free flow of information and knowledge. Those who own or control coding systems have an advantage in regulating the flow of information. It follows that addressing needs for governance in the cyber age must encompass all software, protocols, technical standards, and rules related to coding.

## 3. Needs for Governance in the Cyber Age

How have the above-mentioned developments impacted the operation of a social system whose principal members are territorially delimited states presumed to have the capacity to manage their internal affairs effectively and to formulate and implement coherent strategies in their interactions with one another? Is the growing prominence of virtual reality, in contrast to physical or material reality, a game changer when it comes to the emergence of needs for governance and the efficacy of strategies for meeting these needs?

There is nothing new about the transformative impacts of technological advances on an international and increasingly global scale. Think of the rise of energy systems based on the combustion of fossil fuels, the mechanization of industry, the development of new forms of transportation (e.g., trains, automobiles, airplanes), and the introduction of new communications systems (e.g., telephones, radio, television, film). These technological forces have brought about far-reaching changes, including the transition from a collection of regional political systems to the globalized system we live in today. Yet, none of these technological forces has produced transformative changes in the character of international society itself. Even the advent of nuclear weapons has failed to disrupt the defining features of this society, understood as an anarchical system whose principal members are sovereign states seeking to maximize relative gains in their interactions with one another. Treated at face value, the Charter of the United Nations, adopted at the close of World War II in 1945, takes some significant steps toward a restructuring of the prevailing global order. However, actual experience over the ensuing 75 years has made it clear that this experiment has failed to bring about a transformative change in international society [19].

What makes the onset of the cyber age a game changer in terms of its consequences for global governance? Of course, states make extensive use of cyberspace in their efforts to maximize relative gains [4]. Consider the dramatic growth of activities on the part of states involving uses of cyberspace to compromise or disable the weapons systems of opponents, operate unmanned weapons, and interfere in the electoral systems of other states [20–22]. In these and a variety of other ways, states are simply adding cyber age technologies to the tools they employ in their dealings with one another.

Nevertheless, the steady growth of cyberspace is increasingly eroding the defining features of the states system. Activities involving virtual reality, in contrast to physical or material reality,

pay little attention to spatial boundaries defined in legal and political terms. In a hyper-connected world, individuals and groups interact with one another without reference to conventional boundaries, and states find it increasingly difficult to control their internal affairs effectively. Non-state actors (including multinational corporations) enjoy increased autonomy, and guiding narratives are no longer defined in terms of the pursuit of national interests.

This does not mean that states will not make a concerted effort to maintain effective control over the activities of their citizens and over their domestic affairs more generally. China's experience in constructing the Great Firewall is instructive [22]. The fundamental objective of this effort, which consumes resources comparable to those expended on the military, is to control the access of Chinese citizens to the resources of cyberspace on a global scale. The result is a dual system featuring a sharp separation between internal and external digital systems. There is no doubt that this effort has been effective to some degree. However, what is striking is how easy it is for those, with an incentive to do so, to circumvent the limitations of the Great Firewall. WeChat, a sophisticated Chinese social media system, works perfectly well on a global basis. Chinese citizens can monitor air pollution in real time on their cellphones. Informative programs quickly go viral, before authorities take them down. Even then, those with access to a VPN can continue to view them. Hundreds of thousands of Chinese students are studying abroad at any given time. More and more affluent Chinese citizens travel abroad or even own property in other countries, where they or their relatives spend significant amounts of time. Any Chinese citizen, with the time and inclination to do so, can gain access to the resources of global cyberspace. Overall, the effectiveness of the most determined effort the world has witnessed to insulate a territorially delimited society from the world of cyberspace has been limited.

The onset of the cyber age also is compromising the ability of territorially-based states to insulate themselves from external interventions. Here, the recent experience of the United States is instructive. As WikiLeaks makes clear, it is increasingly difficult to maintain control over information, including highly classified information subject to stringent control procedures. As growing evidence regarding interference in electoral processes demonstrates, states also are finding it difficult to ensure the integrity of their most important political institutions. Attention has focused on the activities of hackers based in Russia and perhaps blessed by the government of Russia, in the case of the 2016 presidential election. However, it is now apparent that unauthorized groups or even individuals hidden in remote locations can interfere in disruptive ways in elections and other domestic political processes of states anywhere in the world.

Similar remarks apply to the capacity of states to implement coherent foreign policies. The case of Stuxnet, in which the government of the United States took steps to disrupt Iran's nuclear program, is prominent in this regard [21]. However, the capacity to engage in disabling cyber attacks is not limited to those acting on behalf of the governments of states. A range of non-state actors can compromise the diplomatic and military activities of states. Moreover, the uncertainty regarding both the nature and the extent of such interference is raising growing concerns about the ability of states to interact with others effectively. Of course, states—especially large, powerful states—are not likely to remain passive in this setting. We are already experiencing an offense-defense race between those seeking to develop disruptive capabilities and those responsible for defending against them. There is no basis today for assuming that the defense will be able to match the rapid pace of development of the offense in this race.

These developments make it clear that the onset of the cyber age is producing profound impacts on the defining elements of the states system and, in the process, calling into question key assumptions about what many refer to as earth system governance and about the constitutive conditions within which the pursuit of national interests plays out [23]. However, this is not the whole story. The cascade of new technological developments is continuously expanding the scope of cyberspace and raising new challenges that must be taken into account as a matter of priority by those interested in meeting needs for governance in today's world. Two clusters of developments are particularly striking in this regard.

One cluster encompasses a suite of advances featuring artificial intelligence, robotics, machine learning, blockchain technology, big data, and the Internet of Things. While states endeavor, with some success, to regulate specific applications of the relevant technologies (e.g., the operation of self-driving vehicles), the variety, scope, and pace of development of these advances makes it impossible for states to maintain effective control over the new world arising from these advances. A particularly notable development is the growth of tighter and tighter linkages between digital systems and material systems, creating a world in which cyber-physical systems are becoming dominant features in many realms, ranging from the use of robots in manufacturing processes and the use of blockchain technology in accounting systems to the development of novel household appliances [14,24]. The challenge of asserting effective control over these developments is daunting, even in a society in which central planning and top-down control are well-established practices, much less in a society in which there is no tradition of central planning regarding developments occurring in the private sector.

The second related cluster centers on growing concerns regarding cyber security [22]. Although states employ cyber weapons themselves [4,23,24], what is striking in this realm is the inability of states to achieve effective control over uses of cyber weapons on the part of others (including single individuals), even when they are located within their own boundaries. In a world of digital-material systems, the capacity of those desiring, for whatever reason, to disrupt the smooth operation of advanced societies is growing rapidly. The possibilities in this realm are endless, ranging from interference in electoral processes, to interventions in financial systems, the disruption of commercial traffic, and the breakdown of medical systems. There are rising concerns about the ability of states to manage their domestic affairs effectively, much less to engage in coherent actions extending beyond their borders.

While the onset of the cyber age itself generates an array of new needs for governance, it also has important consequences relating to broader efforts to address a range of issues often associated with the concept of sustainable development. Sometimes, the results are helpful. With regard to arms control agreements, where issues relating to monitoring, reporting, and verification are critical, the technologies of the cyber age may help to buttress mutual assurance. Similarly, new developments in the realm of information technology may reduce needs for personal travel and for the shipment of material goods, contributing to efforts to limit emissions of greenhouse gases. On the other hand, the technologies of the cyber age may combine with rapid advances in the realm of biotechnology [25] to complicate efforts to address needs for governance relating to matters like human genome editing and bio-electronics.

Connections of the above-mentioned sort open up an array of issues relating to what the United Nations calls "transforming our world" in its 2030 Agenda for Sustainable Development [26]. The key point here is that the onset of the cyber age is occurring in parallel with advances in other fields and that linkages between and among these developments are unavoidable. It follows that efforts to pursue the full suite of goals included in the Sustainable Development Goals must find ways to come to terms with the profound changes affecting efforts to address needs for governance associated with the onset of the cyber age. Achieving a lasting sustainability transition will require a transformation of the prevailing global order.

## 4. The Global Order of the Future

If it is impossible to meet prominent cyber age needs for governance within a system of global order whose centerpiece is a society of territorially delimited states competing for influence in the absence of any overarching public authority, what is the alternative? Can we envision a new order yielding better outcomes? What are the processes that can produce a transition from the prevailing order to such an alternative? Can the digital systems of the cyber age play a role in energizing these processes?

It is important to avoid naivete in thinking about this subject. Inertia or path dependence is a powerful force in human affairs. Nevertheless, we can identify processes of change in key elements of the prevailing order energized, in part at least, by the operation of the digital systems of the cyber age.

Considering both Western thinking and non-Western thinking about mechanisms of social change, we introduce analytic distinctions among three types of change. Our assessment does not yield specific predictions regarding the character of the global order of the future. But it does draw attention to key drivers of largescale social change and demonstrate that the prospect of far-reaching change is not far-fetched.

The institutions of the current global order are social constructs. They have not been with us forever, and they will change over the course of time [27]. What we think of as the Westphalian system, for example, dates only from the middle of the 17th century [28]. The arrangements in place prior to that time differed fundamentally from our familiar system of place-based sovereign states. Moreover, the Westphalian system has evolved substantially over time as a result of modernization, decolonization, globalization, and the rise of a multiplicity of non-state actors [29].

### 4.1. Western Approaches to Global Social Change

Mainstream thinking about the future of the global order is dominated by Western and, more often than not American, thinking about international relations [30]. Much Western thinking directs attention to the character of the major actors and interactions among them. Rooted in neo-realist premises, the resultant work explores the rise and fall of major states, including the fate of the United States as an aging hegemon and the prospects for the emergence of a new balance of power [1,31]. Still, not all Western thinking conforms to this pattern. Two distinct streams of thought are of interest to those contemplating prospects for changes in the prevailing global order needed to meet cyber age needs for governance: (i) a reformist stream emphasizing adjustments that do not envision wholesale or sudden changes in the dominant role of states and (ii) a more radical stream focusing on the prospect of the collapse of the states system.

#### 4.1.1. Facilitated Reform

We are not witnessing the "end of history," an idea arising with the waning of the Cold War and envisioning the spread of liberal democracy and regulated capitalism on a global basis [32]. Political and economic systems in places like China are not converging toward Western models. The gap between the ideal and the actual with regard to liberal democracy is growing, even in the United States and Europe. This rules out any expectation of progressive evolution toward a Kantian world in which the spread of republican or democratic forms of governance among the members of international society will prevent violent conflict and produce what some have called a democratic peace [33]. Nevertheless, this does not mean the global order is static, so that we are doomed to live indefinitely in a society of sovereign states competing to maximize relative gains [29]. Three developments, driven in part by the growing role of digital systems, are likely to have important implications for the treatment of needs for governance arising in the cyber age: (i) social changes occurring within states, (ii) acknowledged restrictions on the sovereignty of states, and (iii) the emergence of various types of non-state actors.

In its pure form, the Westphalian model treats states as billiard balls in the sense that they are hard-shelled units interacting with one another in efforts to maximize relative gains. Analyses based on this premise direct attention to geopolitical processes in which states compete with one another for influence, regardless of the character of their domestic economic, political, and social systems. No state can afford to depart from this mode of operation, lest it be exploited and subordinated to others that do engage in power politics [31]. Yet, this model is simplistic. International society now includes almost four times as many members as it did at the close of World War II. What began as a European invention to solve intractable social problems arising on a regional scale has expanded to global proportions; the diversity among the members of international society has increased. States differ not only in terms of size, population, and resource endowments but also in terms of ethnic composition, culture, economic development, and political arrangements. There is no reason to assume that pressures associated with participation in international society will force convergence regarding the internal characteristics of the members of this society.

It is pertinent to ask whether the members of international society are experiencing forms of social change generating significant consequences for efforts to meet cyber age needs. In an increasingly wired world, the leaders of states are under growing pressure to (at least appear to) pay attention to the social welfare of those residing within their jurisdictions. This does not rule out the rise of authoritarian practices or the growth of socioeconomic inequality. However, it does mean that there is a need to respond to the concerns of members of the public linked via social media. One response to this concern, exemplified by recent developments in the United States, is to adopt an inward-looking stance, endeavoring to protect citizens from external forces seen as detrimental to their welfare.

Due to the growth of the internet, social media, foreign travel, migration, the cross-border movement of goods and services, and the operation of transnational financial flows, however, it is increasingly difficult for leaders to insulate individual states from participation in global systems. Efforts to erect barriers to these linkages, including the Great Firewall, limitations on immigration, trade restrictions, and so forth, can prove at least partially effective under some conditions. Over time, however, domestic policies that depend on an ability to insulate a country from global connections are doomed to failure.

Taken together, these observations give the individual members of international society incentives to come to terms with cyber age needs for governance. The result is a pattern of mixed-motive interactions [34]. States continue to be motivated by self-interest. However, there is growing understanding that the pursuit of self-interest requires cooperative efforts to come to terms with issues like the control of weapons of mass destruction, the concerns associated with cyber security, the impacts of advances in artificial intelligence, and the global spread of contagious diseases. There is no assurance that specific leaders in key states will grasp the implications of embracing enlightened self-interest. Nevertheless, framing the issue in these terms provides some basis for optimism.

These observations lead to a consideration of the second element of facilitated reform. Many assume that sovereignty construed as the right of states to refuse to be bound by international obligations without their explicit consent is indivisible. Yet, this premise, too, is a social construct. In most domestic systems, individuals are obligated to comply with a variety of societal rules in their dealings with others, even though they are free to make their own decisions about how to proceed in many areas. Similar restrictions on the exercise of sovereignty on the part of the members of international society may arise.

Are we moving toward a situation featuring growing restrictions on the exercise of state sovereignty? There is no simple answer to this question. We lack metrics that would allow us to track trends in this realm objectively. Even so, there are indications that states are experiencing growing pressures to conform to international standards articulated through media beyond the control of political leaders. Perhaps the most striking examples involve human rights and measures relating to the protection of the environment.

Individual states continue to behave in ways that are unacceptable in terms of modern perspectives on human rights. Nevertheless, as the operation of bodies like the European Court of Human Rights and the International Criminal Court attest, international concern about violations of human rights within member states is rising [35]. Similarly, the idea of humanitarian intervention has emerged as a justification for outsiders to take actions affecting the domestic affairs of members of international society. These may seem like small steps. Still, they reflect a trend toward the imposition of societal standards on the actions of states within their own jurisdictions.

Similar observations are in order regarding environmental issues like climate change where the domestic actions of states have systemic consequences affecting the welfare of those located elsewhere. The picture is not clear in this regard. Russia has staked its economic revival on the extraction and marketing of fossil fuels. The current administration in the United States has intervened heavily to prop up the coal industry. The government of Brazil is promoting policies leading to accelerated destruction of forests in the Amazon Basin. Nevertheless, we are witnessing the rise of environmental principles that are reflected in the provisions of international legally binding instruments and that

states are finding it increasingly difficult to ignore. This development underlies the generally successful campaign to eliminate ozone-depleting substances [36]. Whether it will prove sufficient to address problems like climate change or the loss of biological diversity remains to be seen.

The answer to this question depends, in part, on a third element of facilitated reform closely associated with the advent of the cyber age: the rise of non-state actors as influential players in international society. Three distinct types of non-state actors are worth distinguishing in this context: intergovernmental organizations (e.g., the UN Environment Programme, the World Bank) that become actors in their own right; multinational corporations and other large economic actors whose influence extends beyond national borders (e.g., Walmart); and non-governmental organizations dealing with matters of social welfare, human rights and the environment (e.g., the Worldwide Fund for Nature).

Non-state actors making use of cyber age technologies and operating beyond the control of national authorities have become influential drivers of the course of world affairs in some important areas [37]. Major corporations control supply chains and trading relationships that are critical to the operation of a globalized economy. Non-governmental organizations have expanded their reach to global proportions in addressing issues relating to health, human rights and the environment. Intergovernmental bodies like the World Trade Organization and the World Bank are capable of acting autonomously, even though they owe their existence to the actions of states.

A particularly notable development is the rise of public-private partnerships. These are initiatives joining together some public entity (e.g., the UN Development Programme, the World Health Organization) and one or more free-standing private entities (e.g., Oxfam, the Gates Foundation) to address some need for governance. Skepticism regarding the efficacy of these partnerships is justified in specific instances. Nevertheless, evidence is growing that they do make a difference in some issue areas that are significant from the perspective of sustainability [38].

### 4.1.2. Managed Transformation

What all these perspectives have in common is that they are reformist in character. They all focus on changes in the character of key actors unfolding over time and assume that developments affecting the nature of the prevailing global order will involve incremental changes. Meeting new needs for governance in the cyber age can occur on a piecemeal basis. This may well be the proper way to think about changes in the prevailing global order. Yet, this reformist approach does not inspire confidence regarding efforts to meet cyber age needs for governance, and it does not address the possibility of transformative change in complex systems. The fact that we are now operating in a world of complex systems whose non-linear dynamics produce changes that are often rapid and frequently surprising means that we cannot rule out the occurrence of transformative change in the defining features of the prevailing global order. The rise of digital systems has heightened the operation of these dynamics. Just as we now think about planetary boundaries in biophysical systems and seek to identify thresholds and trigger mechanisms leading to bifurcations, we can ask about similar phenomena occurring in the institutional foundations of the existing global order. The fact that the principal elements of international society are social constructs suggests that they may be particularly susceptible to transformative change [39].

The occurrence of bifurcations in large social systems is difficult to anticipate. Even in cases where conditions leading to transformative changes build up over time, we are regularly taken by surprise when explosions occur. Think of cases like the onset of World War I in 1914, the start of the Great Depression in 1929, and the collapse of the Soviet Union in 1991. In all these cases, close observers could tell that the status quo was unstable, so that crossing a threshold leading to transformative change was a distinct possibility. Yet, in each case, highly knowledgeable observers were taken by surprise when disruptive change occurred. As the scope of the systemic disruptions arising from COVID-19 suggests, the prospect of similar surprises occurring on a global basis is heightened in the digital age.

This suggests two significant observations relating to managed transformation. One involves the importance of engaging in systematic thinking in advance regarding alternative arrangements. Transformative changes in social institutions can produce windows of opportunity in which managed transformation becomes possible. Institutional changes that would have seemed fanciful prior to the transformative moment suddenly become possible; the operation of digital systems allows new narratives to diffuse with remarkable speed [40]. However, these windows do not remain open indefinitely. This makes it important to devote resources to thinking carefully about alternative arrangements in advance. This can be a frustrating exercise; opportunities to move the results from paper to practice may never arise. Still, running the risk of irrelevance is preferable to being unprepared when an opportunity does arise.

This leads to the second observation. Failure to prepare adequately for opportunities of this sort can lead to periods of chaos and eventual relapse into socially undesirable arrangements. The operation of digital systems can intensify these problems. The political system emerging in Russia following the collapse of the Soviet Union failed to provide effective barriers to the reemergence of authoritarian rule. Adjustments introduced following the great recession of 2008–2009 are insufficient to offer any guarantee against a recurrence. Similar observations apply to transformative changes in the prevailing global order. While planning for an uncertain future is difficult, assuming that things will work out for the best once a transformation occurs is a recipe for failure, especially in addressing broad concerns relating to sustainability. Today, we face the challenge of avoiding a return to the status quo ante regarding the threat of destructive pandemics once the COVID-19 crisis subsides.

### 4.2. Chinese Approaches to Global Social Change

As the dominance of the United States wanes, it is increasingly apparent that there are alternative streams of thought about global order that introduce different perspectives relevant to meeting cyber age needs for governance. The dramatic rise of China has triggered a sharp growth of interest in Chinese perspectives on largescale social change and how it can occur [30,41,42].

Chinese thinking about such matters stretches back over several millennia. What stands out is an emphasis on the role of ideas, including moral and ethical principles, as determinants of social order and the premise that it is possible to steer human behavior through education in contrast to the development of regulatory measures and effective compliance mechanisms. Chinese thinking does not highlight the role of states (much less nation states) and does not take it for granted that international society is an anarchic society of sovereign units. The basic building blocks are not place-based states that compete with one another to maximize relative gains. Interactions among socially organized units can reflect common worldviews and feature economic and sociocultural exchanges, producing win-win results in contrast to geopolitical interactions centered on the struggle to establish hierarchical relationships based on the distribution of political power.

Digital systems play a powerful role in facilitating the rapid spread of ideas. As recent developments in China make clear, public authorities can endeavor to use these systems to control the thinking of members of the general public. However, the spread of ideas in the cyber age has a dynamic of its own that can lead to sharp shifts in thinking over short periods of time that are difficult to control.

#### 4.2.1. *Tianxia* and *Gongsheng*

Perhaps the most significant efforts to distill Chinese ideas relevant to the future of the global order are associated with the concept of *tianxia* [41]. The term *tianxia*, literally all under heaven, is a label used to describe a system of relationships arising during the Zhou Dynasty in which a constellation of actors remain in orbit around a central node due to the effects of centripetal forces involving cultural practices and economic factors more than political forces. In this system, "China resided in the center and became the suzerain state," creating a whirlpool effect in which many actors (sometimes called vassal states) paid tribute to the central actor, while that actor devoted more attention to maintaining the resilience of the system than to aggrandizing itself [43] (p. 32).



In this system, a process known as recomposition, similar to genetic recombination, served to produce an integrated system. This process was not a matter of imposition of norms and rules on the part of a hegemon but "an interactive process involving multiple parties" [43] (p. 33). The result was a resilient system "responsive to changes and adaptable to new circumstances" [43] (p. 33). The hallmark of this form of order is the idea that it is possible to achieve order through a system in which loyalty to a central actor (the emperor in classical Chinese thinking) committed to promoting the common good can serve as a basis for promoting harmony in interactions among a diverse collection of social units.

The critical question today is whether the *tianxia* system can be broadened from an Asian application to serve as a basis for a global order conducive to meeting cyber age needs for governance. The answer to this question is not straightforward. What is needed is a "realization that we have reached a critical juncture in history" [43] (p. 45), a development requiring the emergence of a global narrative replacing the narrative of the Westphalian order with a communitarian narrative highlighting the need for cooperation as a basis for enhancing human welfare. Key elements of such a narrative include the idea of relational rationality in contrast to individual rationality, Confucian improvement as an alternative to Pareto optimality, and universalism based on a commitment to peaceful coexistence.

*Tianxia* acknowledges that different types and sizes of social units under heaven exist simultaneously. However, the concept avoids dealing with them separately or in isolation. It emphasizes holistic and communitarian responses to problems. Under *tianxia*, the focus is on reconciling contradictions between intertwined areas in order to avoid overall imbalances. Chinese philosophy highlights grayscale thinking (taiji). We should find the roots of systemic problems instead of treating the symptoms. In the cyber age, the frequency and intensity of interactions among actors are increasing continuously; boundaries are becoming blurry. The Chinese approach emphasizes the balance and resilience of systems themselves as the key to meeting needs for governance rather than seeking to impose clear-cut boundaries.

Another approach, known as *gongsheng* theory, seeks to answer the question of how to promote coexistence and limit conflict in an international society comprised of "numerous states and non-state actors, who diverge, sometimes quite considerably, in size, wealth, value and tradition" [44] (p. 3) The key idea involves social symbiosis along with the insight that differences among social units may provide the basis for developing mutually beneficial relationships, contributing to sustainability in the process.

Proponents of the concept of *gongsheng* reject the Social Darwinian "belief that nature is a brutal war of all against all" and therefore, the focus on actors seeking to maximize relative gains [44] (p. 7). Rather, "under a condition of pluralism, homogeneous and heterogeneous things mutually accommodate and mainly complement each other. They appreciate and inspire each other and grow together," which is an idea with a long tradition in Chinese thought [44] (p. 9).

This perspective, too, may seem fanciful to Westerners. Nevertheless, like *tianxia,* the idea of *gongsheng* raises questions about the fundamental premises of Western thinking regarding the institutional dimensions of global order and the role of governance. It forces us to reflect on Western assumptions about the sources of human behavior that may turn out to be part of the problem rather than part of the solution in meeting cyber age needs for governance.

### 4.2.2. Applying Chinese Thinking

The meaning of some Chinese terms relating to social order differs from their Western counterparts. For example, Chinese speakers use the word *anquan* to translate the English word security. This translation reflects the idea of ensuring that no adverse changes occur. However, the word *anquan* in Chinese also includes other meanings, such as peace, stability, comprehensive and all together, not normally associated with the English word security. Chinese thinking reminds us that there are different ways to conceptualize concepts like order and sustainability in human affairs. Some alternatives place more emphasis on economic and sociocultural benefits to be derived from peaceful coexistence than on the human desire to reach the top in the struggle for relative gains.

Chinese history features a significant gap between the ideal and the actual in these terms. Periods of disorder marked by largescale violence, for example, are by no means unusual in the East Asian cultural zone. Many also see Chinese ideas like *tianxia* as no more than a gloss on imperialistic efforts to exercise power. Nevertheless, these Chinese perspectives on global order make us stop to reassess premises about impediments to addressing needs for governance that are not based on unchanging laws of nature. In this connection, it is important to bear in mind that cyber age technologies allow new ideas to spread rapidly on a global basis in ways that are difficult for states to control.

With regard to meeting cyber age needs for governance, one important idea arising from China's experience involves an approach to governance that we now characterize as goal-based governance in contrast to mainstream Western thinking emphasizing rule-based governance [45,46]. The Chinese government establishes medium- and long-term goals in Five-Year Plans and launches campaigns to fulfill them. Regular assessments of progress toward the achievement of these goals lead to periodic adjustments in the deployment of resources. This approach has helped China achieve rapid economic and social development. It has obvious relevance at the global level in such forms as the campaign to fulfill the Sustainable Development Goals.

## 5. Cyber Age Governance in Context

What can we conclude from this analysis about the prospects for meeting cyber age needs for governance? The argument of the preceding section highlights processes of change in the prevailing global order energized in part by the operation of the digital systems of the cyber age themselves. However, it would be premature to make specific predictions about the character of the global order of the future, much less about the prospects for developing effective ways to address cyber age needs for governance in this setting.

Largescale changes in complex social systems feature non-linear processes and often surprising outcomes. The flow of technological developments of the sort that have given rise to the cyber age will continue and may well accelerate. These developments will interact in a variety of ways with advances in other areas, such as biotechnology and neuroscience.

Nevertheless, we see two critical issues relating to the global order of the future that are likely to play a determinative role regarding the fate of efforts to meet novel needs for governance arising in the cyber age. One centers on the extent to which place-based states driven by a desire to maximize relative gains continue to constitute the dominant actors in earth system governance. The other concerns the prospect that the operation of digital technologies will promote the spread of a new narrative featuring a communitarian vision and the importance of collaborative approaches to meeting cyber age needs for governance.

States competing with one another to maximize relative gains will seek to make use of digital technologies as tools for exercising power, while working to defend themselves from intervention on the part of outsiders, ranging from competing states down to the level of individuals capable of deploying digital technologies covertly. This will intensify novel needs for governance rather than creating conditions under which new approaches to governance can flourish. Those who think in neo-realist terms may take it for granted that there is no alternative to a world of competing states. Yet, as the discussion of facilitated reform and managed transformation in the preceding section suggests, there are processes, energized in part by the operation of digital technologies, that can lead to far-reaching change in this feature of the prevailing global order.

Nor are ideational processes of the sort emphasized in Chinese thinking about global social change far-fetched. Cyber age technologies, including various types of social media, make it possible for new social narratives to spread globally almost overnight. It is true, of course, that states can and do seek to control the flow of information and to make use of social media in their struggle to maximize relative gains. Recent experiences with outside intervention in domestic electoral processes provide dramatic examples. Yet, the ability of states to control the emergence of new social narratives is limited. The case of China is instructive in this regard. The massive effort put into the development of the Great Firewall

is effective to a degree. Still, it may prove impossible to prevent the spread of the communitarian vision embedded in the concepts of *tianxia* and *gongsheng* on a global scale.

To put this analysis into a broader context, we ask whether our assessment of the challenge of meeting cyber age needs for governance has implications for efforts to fulfill the vision articulated in the 2012 Declaration of the UN Conference on Sustainable Development entitled "The Future We Want" [47] and for the pursuit of the UN's 2030 Sustainable Development Agenda launched in 2015 as an exercise in goal-based governance designed to move this vision from paper to practice [26]? We believe it does. For one thing, the development of a capacity to meet the novel needs for governance arising in the cyber age is itself a critical component of the larger effort to move toward the future we want. A world preoccupied with the challenges of cyber security from the individual to the global level does not lend itself to efforts to make progress regarding the familiar pillars of sustainable development dealing with social well-being, economic prosperity, and ecological integrity.

What is more, our discussion of the importance of alterations in the character of the prevailing global order dovetails with recent thinking regarding the importance of transformation as a condition for fulfilling the Sustainable Development Goals [48]. We acknowledge that it is possible to make significant progress toward fulfilling goals like reducing poverty within the confines of the existing global order, though it is important to recognize that most progress regarding this goal stems from the effects of economic development in countries like China rather than from initiatives launched under the auspices of the Millennium Development Goals prior to 2015 and the Sustainable Development Goals beginning in 2016. When it comes to fulfilling more systemic goals relating to matters like climate change, the loss of biodiversity, and rising threats to the oceans, however, it is difficult to see a pathway to success within the confines of the prevailing global order. States competing to maximize relative gains have little capacity to solve collective action problems of the sort underlying the climate emergency or to address externalities of the sort leading to the loss of species or the spread of plastic debris in the oceans. What is needed to address these issues effectively is transformative change in the prevailing global order. There is no guarantee that we will succeed in fulfilling this condition in time to avoid costly disruptions in many realms. Nevertheless, our discussion of meeting cyber age needs for governance suggests that critical transitions of this sort do occur and that the operation of the digital systems of the cyber age may play a central role in driving such transitions [39].

**Author Contributions:** O.R.Y., J.Y., and D.G. developed the argument presented in this article. O.R.Y. and J.Y. prepared the original version submitted to *Sustainability.* O.R.Y. and D.G. revised the argument in response to the reviewers' comments. O.R.Y. prepared the revised version for submission. All authors have read and agreed to the published version of the manuscript.

**Funding:** This research received no external funding.

**Conflicts of Interest:** The authors declare no conflict of interest.

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
