# Peer review of "Meeting Cyber Age Needs for Governance in a Changing Global Order"

_sustainability, doi:10.3390/su12145557_

Round 1

Reviewer 1 Report

The article is coherent, clear and highly readable presenting an important reflection on the social and political implications that the advent of the cyber age may have “for the pursuit of sustainability in a changing global order” (20-21). It is likely to attract a wide readership and it sets an agenda for further research in the fields of global politics and international relations. I support the publication of this work in due time and I am looking forward to read the printed version. Nevertheless, I have some suggestions, which I hope may be useful to further clarify and expand some key elements of the argument presented.

My main concerns regard the following aspects: 1) the main claim of the article is not immediately clear; 2) the structure of the argument risk being confusing; 3) the relation between sustainability and order is left under-theorized 4) parts of the argument may be controversial and they may need further elaboration and clarification. Let me explain what I mean by each of these points:

First of all, the main claim of the article is not immediately clear. The abstract promises that the article will “investigate the implications of this development [the advent of the cyber age] for the pursuit of sustainability in a changing global order”. And yet, after reading the contribution, it is not immediately clear to me what these implications are. The first part of the article convincingly argues that “the onset of the cyber age has compromised the capacity of states to respond to pressing needs for governance”. On the other hand, the second part focuses on a different set of transformations that “facilitate transformation” and develops a summary view of “Chinese approaches to global social change”. In the second part, the focus on the cyberage appears to recede in the background and instead the article focuses much more generally on future perspectives for global governance. This is not necessarily a major problem but it would have been really helpful if the second part included a more direct discussion of how the advent of the cyberage impact future perspectives for global governance. For instance, the article argues that three “types of change likely to have important implications for the treatment of needs for governance and the pursuit of sustainability in the cyber age are worthy of consideration: (i) social changes occurring within states, (ii) acknowledged restrictions on the sovereignty of states, and (iii) the emergence of various types of nonstate actors” (376-380). These transformations are treated as parallel with and rather independent from the “advent of the cyberage”. I wonder, nevertheless, if the article could have emphasized the way in which these transformations intertwine with “the advent of the cyberage”? For instance, how has the development of cybertechnologies impacted social change occurring within states? Has it facilitated or impaired the development of restrictions on the sovereignty of states? Has it accelerated or slowed down the emergence of powerful non-state actors?

These questions lead me directly to my second concern. The structure of the argument – which is divided in five sections, two subsections and four sub-sub-sections – is not immediately clear. In particular, there seems to be a caesura between the first three sections and the fourth section. This latter section is focused on sketching alternatives to the state system that may be able to “meet key needs for governance arising in the cyber age in such a way as to promote sustainability” (333-334). And yet, the section does not clarify explicitly how the two ‘alternatives’ presented in subsections 4.1 and 4.2 may be able to “meet key needs for governance arising in the cyber age in such a way as to promote sustainability”. For instance, how does the “the rise of nonstate actors as influential players in international society” may be able to “meet key needs for governance arising in the cyber age”? Or, similarly, how may a turn to Chinese thinking concerning global change contribute to address the specific problems arising from the “advent of the cyberage”? The answers to these questions are left implicit and it is not fully developed. I wonder if it would be possible to make these issues more explicit. This would also contribute to link more directly the fourth section with the three preceding sections.

My third concern relates to the use of the concept of “sustainability” in the article. The concept is invoked several times, but it is never defined and there is not explicit discussion of the theoretical controversies surrounding its meaning. In particular, it sometimes seems that the article reduces “sustainability” to “order” and “governance”. And yet, a “stable”, “orderly” and thoroughly “governed” global system may still perpetrate an “unsustainable” socio-ecological metabolism. In other words, is “unsustainability” simply the result of “insufficient governance”? And, on the other hand, does more international cooperation - and global governance - result necessarily in a more sustainable mode of production? How does the “advent of the cyberage” impact the socio-ecological metabolism? Does it lead to a more or less ‘sustainable’ international system? I would have liked the article to consider more explicitly these issues.

Finally, I have a number of questions concerning specific passages in the paper that may need further clarification and/or development:

a) At 102 the article states that “increasingly digital systems interact with both biophysical and social systems in transformative ways”. And yet, I wonder if it is useful to think of ‘digital systems’ as even partially independent from biophysical and social system (as represented in Figure 1)? Aren't digital systems just a sub-division of "social systems"? Aren't they a ‘social product’ and very material infrastructure including energy-intensive – and environmentally damaging - server farms? In other words, is there an element of "digital systems" that is not social? How do we make sense of the space that is part of digital systems and yet outside of "social systems" (Figure 1a)? And at the same time, are not digital systems always biophysical? In short, wouldn’t it be more correct to think of “digital systems” as a sub-division of “social systems” (and of “social systems” as a subdivision of “biophysical systems”) [for this alternative conception, see for instance: Figure 1 in Capra, F., & Jakobsen, O. D. (2017). A conceptual framework for ecological economics based on systemic principles of life. International Journal of Social Economics, 44(6), pp. 831-844. In their perspective, the “economy is a living system nested in other living systems – society, culture, politics, nature, and ultimately Gaia, the living Earth (836)]?

  1. b) At 148-149, the article states that “hysteresis produces lags in the manifestation” of the social impact of digital technologies, so that we “can expect the social impacts of today’s technological innovations (e.g., 5G, AI, 149 human genome editing) to lag 5-10 years behind the invention of the technologies themselves”. I find problematic the use of the concept of “hysteresis” in this passage. I may be erroneous but the concept is usually employed to indicate the phenomenon by which the value of a physical property lags behind changes in the effect causing it. Hysteresis can be observed in physical, chemical and socio-economic systems, but it is not a ‘law of nature’ that can produce lags. In other words, my understanding is that the “lags in manifestation” that the article refers to may well be examples of hysteresis, but they can not be “produced by hysteresis”. In any case, I think the passage may require further clarification. For instance, I wonder why the authors are confident that “we can expect social impact of today’s technological innovation to lag 5-10 years behind the invention” rather than 10-20 or 20-40 years? I am not claiming that the statement is incorrect or that it needs to be removed but I do think that it may arise doubts in the readers; thus, it would be useful if it was further clarified.

  1. c) At 322-324, the article states that  “the technologies of the cyber age may combine with rapid advances in the realm of biotechnology to complicate efforts to address governance issues relating to matters like human genome editing and bio-electronics”. It is an important point. Nevertheless, I am surprised that the reference for this statement is: “Church, G. and Regis, E. Regenesis: How Synthetic Biology Will Reinvent Nature and Ourselves; Basic Books: New York, NY, USA, 2012”. I do not remember the book making this claim and, unfortunately, the reference does not indicate a precise page. Could you please add this information?

  1. d) At 448-453, the article claims that the rise of "nonstate actors" represent "an element of facilitated reform". This is a dubious claim that would require a more extended discussion. One could as easily argue that the growing power of multinational corporations make it more (rather than less) difficult to reform the global system and achieve “sustainability” (for instance, many have argued that the growing power of multinational corporations and the hegemonic voice of NGOs such as the World Economic Forum have contributed to further neoliberal policies that have caused an increase in inequality and slowed down solutions to the ecological crisis. For example see: Harvey, D. (2007). A brief history of neoliberalism. Oxford University Press, USA; or Fuchs, D. A. (2007). Business power in global governance. Boulder, CO: Lynne Rienner).

Finally, I take the opportunity to remark that the article presents a clear, significant and important contribution to international debates on international governance. I hope my comments will not be appear as a sterile critique but rather as a genuine attempt to provide the authors with some sympathetic suggestions.

Author Response

Response to Reviewer 1

We appreciate the overall positive assessment of Reviewer 1. Even more important, we want to express our gratitude to this reviewer for reading our manuscript carefully and raising a range of thought-provoking issues that call for significant revisions in the text. Rather than attempting to address these concerns with minimal or cosmetic adjustments, we have chosen to invest the time and energy required to introduce more substantial revisions in the text.

We are not sure that the new version of the text will satisfy all of Reviewer 1’s concerns. But we have made extensive changes to address the principal issues the reviewer raises. We think the revised text is considerably stronger as a result and hope the reviewer will agree.

1. We have revised the manuscript to highlight the following claims. Mainstream debates about the future of the postwar global order deflect attention from developments that are more fundamental than shifts in relationships among powerful states. The onset of what we call the cyber age has generated new needs for governance that are difficulty to address within the confines of the prevailing global order in which place-based states compete to maximize relative gains. This prompts an analysis of processes of change in this order. We distinguish several processes energized in part at least by the growing role of the digital systems of the cyber age. Whether the global order of the future will provide effective mechanisms to meet cyber-age needs for governance, much less to fulfill the goals of the UN’s 2030 Agenda for Sustainable Development, remains to be seen.

2. We think the revisions we have made also help to make the structure of the argument clearer. We proceed in four steps. The first step introduces the cyber age and considers its transformative impact. In step two, we argue that the onset of the cyber age has led to the rise of novel needs for governance difficult to address within the confines of the existing global order. This leads to step three in which we focus on processes of change in this order energized in part by the operation of digital systems. The final step seeks to put the argument in context, asking about the prospects for meeting cyber-age needs for governance and, more broadly, the UN’s Sustainable Development Goals in a changing global order.

3. We agree fully with the reviewer’s comments about the concept of sustainability. We have revised the text to make it clear that our focus is on meeting cyber-age needs for governance. Thus, we avoid ambiguous references to sustainability in the main sections of the manuscript. Yet there is an important link between addressing cyber-age needs for governance and fulfilling the objectives articulated in the UN’s Sustainable Development Goals. In the concluding section, we endeavor to make this link explicit.

4a. Distinctions between or among systems are analytic in character. Their purpose is to draw attention to important phenomena, even when the systems interact with one another. For example, analysts distinguish between ecosystems and human systems, though we know they interact. The same is true of distinctions between economic systems and political systems. The key point for us concerns the distinction between digital systems and material or physical systems. We have revised our text to highlight differences between material or physical systems and digital systems and to draw attention to the transformative impacts of the rise of digital systems. Having done so, we also consider interactions between the two types of systems. In this regard, we follow the recent work of Laura Denardis on the interactions of cyber and physical systems.

4b. Because there is considerable ambiguity regarding uses of the term “hysteresis” among those who analyze various types of systems, we have dropped the term in the revised version of the manuscript, making it clear that our concern is with lags between causes and the manifestation of effects.

4c. Our purpose in citing the Church and Regis book is to provide a convenient overview of “rapid advances in the realm of biotechnology.” The book does not make claims about the consequences of developments in biotechnology for efforts to meet needs for governance, although it does suggest that once technologies arise, people will find ways to use them in pursuit of their interests. We have relocated the citation to avoid any confusion about this matter.

4d. We agree with the reviewer’s observation that the actions of nonstate actors may complicate efforts to achieve “sustainability.” Our point is that the rise of nonstate actors, energized in part by the operation of digital systems, is one factor leading to change in a global order dominated by states presumed to be capable of managing their internal affairs effectively and protecting themselves from the interventions of outsiders. We hope our revisions regarding the whole issue of sustainability will serve to avoid any misunderstanding regarding this matter.

5. We have revised the title of the paper to convey a clearer sense of the central focus of our analysis.

We hope that we have understood Reviewer 1’s observations correctly and that our revisions help to clarify and sharpen our central argument. We note this exchange could provide the basis for a more extensive engagement regarding a range of important topics and hope to be able to continue working on these matters in the future.

Reviewer 2 Report

This is an interesting paper on the new global order and the cyber-age.  I was expecting more discussion on the role of corporations in this new governance paradigm. Corporations, be they private, public or state owned enterprises are foundational actors in this new global order. The authors need to delve into this more. 

Another missing point missing from the manuscript is the development of blockchain technology to bring transparency into global governance. The article by Nikolakis, John and Krishnan (2018), provides insight into how blockchain can bring transparency into a globalized and fragmented world: Nikolakis, W., John, L., & Krishnan, H. (2018). How blockchain can shape sustainable global value chains: an evidence, verifiability, and enforceability (EVE) framework. Sustainability, 10(11), 3926. 

Author Response

Response to Reviewer 2

We appreciate Reviewer 2’s observation that this is “an interesting paper.”

Reviewer 2 raises two substantive concerns:

1. We agree that the role of non-state actors, including but not limited to corporations, is a major topic to be considered in thinking about the future of the global order. In our analysis, we have directed attention to processes, energized in part by the operation of digital systems, that are likely to initiate major changes in a global order dominated by interactions among sovereign states seeking to maximize relative gains. To the extent that changes leading to an attenuation of the dominant role of states do occur during the foreseeable future, it will become important to pay more attention to a variety of non-state actors and their efforts to make use of digital technologies to pursue their own interests. We regard this as an important topic for the next phase of research.

2. We agree that blockchain technology is important, and we have added it to our list of digital technologies that have arisen in the cyber age. Among other things, blockchain technology offers the prospect of creating cryptocurrencies and digital accounting systems that operate outside the control of states and that non-state actors can use for a variety of purposes. This is obviously a complex development, but we are not at all sure that it will “bring transparency into global governance.” We do not analyze the operation of any specific digital technology (e.g. AI, robotics, machine learning) in detail in this paper. Our goal is to provide a general account of cyber-age needs for governance and processes of change in the prevailing global order that may affect efforts to meet these needs in the future.